# Native Bamboo (*Indosasa shibataeoides* McClure) Invasion of Broadleaved Forests Promotes Soil Organic Carbon Sequestration in South China Karst

**Zedong Chen** [1,2,3], **Xiangyang Xu** [1], **Zhizhuang Wu** [1], **Zhiyuan Huang** [1], **Guibin Gao** [1], **Jie Zhang** [1] **and Xiaoping Zhang** [1,*]

1   China National Bamboo Research Center, Key Laboratory of State Forestry and Grassland Administration on Bamboo Forest Ecology and Resource Utilization, Hangzhou 310012, China; zedongchen@caf.ac.cn (Z.C.); xuxiangyang202210@163.com (X.X.); wzzcaf@126.com (Z.W.); zhiyuanhuang@caf.ac.cn (Z.H.); anshu998@163.com (G.G.); zhangjie@caf.ac.cn (J.Z.)
2   National Long-Term Observation and Research Station for Forest Ecosystem in Hangzhou-Jiaxing-Huzhou Plain, Hangzhou 310012, China
3   College of Biology and the Environment, Nanjing Forestry University, Nanjing 210037, China
*   Correspondence: xiaopingzhang@caf.ac.cn

**Abstract:** Bamboo invasion into broadleaf forests is a common phenomenon in karst areas; however, the effect of bamboo invasion on soil organic carbon (SOC) in karst areas and the mechanism of the effect are not clear. We selected the study site with broad-leaved forests (BF), mixed forests (MF), and pure bamboo (*Indosasa shibataeoides* McClure) forests (IF). Furthermore, we sampled the soil from 0 cm to 20 cm and 20 cm to 40 cm layers in the region and investigated the soil properties, organic carbon fractions, and microbial communities. At the same time, we sampled the litterfall layer of different stands and determined the biomass. The results showed that bamboo invasion increased the litterfall biomass per unit area of karst forest, increased the bulk weight of the 0–20 cm soil layer, and lowered the soil pH in the 0–20 cm and 20–40 cm soil layers, bamboo invasion consistently increased the content of soil AN and AK, whereas the content of AP was significantly reduced after bamboo invasion. Both active organic carbon groups (MBC, DOC, and EOC) and passive organic carbon groups (Fe/Al-SOC and Ca-SOC) increased significantly after bamboo invasion. The bamboo invasion increased the diversity of soil microorganisms and bacterial communities; the relative abundance of Actinobacteriota increased in MF and IF, while the relative abundance of Firmicutes decreased in IF. The structure of fungal communities was altered during the bamboo invasion, with an increase in the relative abundance of Mortierellomycota and a decrease in the relative abundance of Basidiomycota at the level of fungal phyla. Partial least squares path modeling analysis identified bamboo invasion enhanced SOC sequestration mainly by increasing litterfall biomass and altering the structure of the fungal community, and the effect of bacteria on SOC was not significant. Our study suggests that bamboo invasion of broadleaf forests is more favorable to soil organic carbon sequestration in karst areas.

**Keywords:** plant invasion; soil organic carbon; karst soils; soil microorganism; litterfall

## 1. Introduction

Bamboo forests are a common forest type in subtropical and temperate regions. China has abundant bamboo forest resources, with a national bamboo forest area of 7,562,700 hm$^2$, accounting for 3.31% of the forest area [1]. Bamboo species, as clonal plants, are highly reproductive, and their whip–root growth and asexual reproduction make the bamboo invasion of broad-leaved evergreen forests a common phenomenon in subtropical forests. Bamboo species continuously invade neighboring communities, and the bamboo forest area shows an invasion trend, resulting in problems such as the loss of biodiversity [2],

difficulties in natural regeneration, and stagnation of community succession [3]. It is widely recognized that bamboo forests have high biomass productivity, short rotation cycles, high ecological value, and strong carbon sequestration capacity [4]. However, the effects and mechanisms of native bamboo invasion into broadleaf forests on SOC are not clear. Karst forests are vegetated ecosystems overlying carbonate rocks and calcareous soils rich in $Ca^{2+}$ and $Mg^{2+}$. And bamboo forests and bamboo-broadleaf mixed forests are common forest types in the Karst region of southwestern China [5].

Soil organic carbon (SOC) content and fraction are importantly related to the forest nutrient cycle and microbial metabolism, among other processes [6,7]. Among the studies of soil carbon in bamboo forests, researchers usually divide SOC into two parts: active and passive carbon pools and different SOC fractions have different roles in soil nutrient turnover and microbial structural transformation [8]. Activated carbon pools are the more active components of SOC and are easily utilized by soil microorganisms [9]. Microbial biomass C (MBC) is the most active fraction in the soil, and it makes up the smallest percentage of the SOC, about 4% [10], but it is an important component of soil nutrients [11]. Soil microorganisms vary in their response to changes in SOC [12]. Some studies have shown that increases in plant diversity and litterfall can enhance the storage of SOC through the proliferation of soil microorganisms [13,14]. In contrast, some studies have shown that an increase in litterfall inputs reduces the efficiency of the microbial communities to decompose organic matter, thereby reducing soil C storage [15]. Overall, microbial communities are deeply involved in the soil carbon cycle in different environments.

Microorganisms are important indicators of temporal and spatial variability in SOC, and SOC inputs and outputs are primarily driven by plant and soil microbes [10]. Many microbial species favor the decomposition of SOM, such as Ascomycota, Basidiomycota, and Proteobacteria, further affecting the composition of C fractions [16]. Litterfall decomposition is the main pathway for microbial carbon acquisition, and the rate of decomposition is limited by N during the early stages of litterfall decomposition, while lignin abundance and composition limit the rate of cellulolytic decomposition during the later stages of litterfall decomposition [17]. Fungi play a key role in lignin degradation using a mixture of oxidizing and hydrolyzing enzymes [18,19]. The most studied phenol oxidase is the fungal laccase; many fungi have a number of laccase sequences that are used to encode intracellular and extracellular enzymes that are able to acquire C and N [20]. Soil peroxidases are mainly supplied by Basidiomycota, although some studies have found that a range of soil bacteria are capable of producing peroxidases [21]. Dehydrogenases, primarily intracellular oxidases, transfer hydride groups from substrates to receptors such as NAD+. They are key players in catabolism, particularly for bacteria, as they are often the final consumers of aromatic compounds [21,22].

Bai bamboo (*Indosasa shibataeoides* McClure) is a lignified bamboo plant belonging to the family Gramineae and the genus Rhizophora that is widely distributed in southern Hunan, Guangdong, and northern Guangxi, China. Bai bamboo could invade neighboring broad-leaved forests, forming the secondunderstory of forest stands. It is a very common phenomenon in karst forests where Bai bamboos are distributed; however, little has been reported on the effects of native bamboo species invading broadleaf forests in karst areas. In this study, we selected three stages of the bamboo invasion in the karst region, investigated litterfall biomass, soil physical and chemical properties, and microbial community, and evaluated the effects of bamboo invasion on soil organic carbon fractions and bacterial and fungal communities. The aims are to (1) determine the effects of native bamboo invasion on pertinent soil properties and microbial communities in karst forests; (2) investigate the effects of native bamboo invasion on changes in SOC and C fractions in the karst region; and (3) elucidate the mechanisms and drivers of native bamboo invasion affecting SOC. These results will help us better understand the connection between forest vegetation and soil, and they will provide a theoretical basis for improving the carbon balance and increasing the forest ecosystem's C storage.

## 2. Materials and Methods

### 2.1. Study Site

Ronglinchong Mountain in the Multinational Autonomous County of Longsheng, Guilin, Guangxi Province, China (110°10′ E, 25°80′ N), was selected as the study region (Table 1). The region experiences a subtropical monsoon climate, with an approximate altitude of 900–1000 m, annual sunshine time of 1670 h, annual rainfall of 1500–2400 mm, an average annual temperature of 18.1 °C, an absolute minimum temperature of 4.8 °C, an absolute maximum temperature of 39.5 °C, and the frost-free period is about 314 days. The study area has a typical karst landscape, and the soils in the area are classified as Haplic Acrisol, according to the WRB (World Reference Base for Soil Resources). The vegetation types found in the region include three transects spanning a broadleaved forest, mixed forest with Bai bamboo growing in the understory of broadleaf trees, and Bai bamboo (*Indosasa shibataeoides* McClure) forest.

**Table 1.** Summary of vegetation characteristics within the different stands at different bamboo invasion stages.

| Stands | Altitude/m | Longitude and Latitude | Aspect and Slope | Bamboo | | | Broadleaved Tree | |
| | | | | DBH (cm) | Height (m) | Density Stem (hm$^2$) | DBH (cm) | Density Stem (hm$^2$) |
|---|---|---|---|---|---|---|---|---|
| BF | 945 | 110°10′ E, 25°80′ N | WS 30° | - | - | - | 9.5 | 2374 |
| MF | 945 | 110°10′ E, 25°80′ N | WS 29° | 24.65 | 6.86 | 3450 | 8.7 | 948 |
| IF | 950 | 110°13′ E, 25°47′ N | WS 30° | 16.03 | 4.59 | 6600 | - | - |

### 2.2. Experimental Stands

Based on our background investigation, all bamboo, broadleaf, and mixed forests in the study area have had little human intervention since the 1990s. The dominant species in the secondary broadleaved forest (BF, representing a stand that has not been invaded by bamboo) were *Adinandra megaphylla*, *Liquidambar formosana*, and *Itea oblonga*. The shrub layer mainly consisted of *Eurya groffii*, *Maesa japonica* (Thunb.), *Oxyspora paniculata*, *Camellia oleifera*, *Lindera glauca*, *Ternstroemia gymnanthera*, and *Ilex viridis* (Table 2). In mixed forests (MF, representing stands where bamboo and broadleaved trees form mixed forests), the upper layer was composed of broadleaved evergreen trees, and the lower layer of broadleaved evergreen trees was composed of Bai bamboo and a relatively small number of shrubs and herbaceous plants. In the invaded Bai bamboo forest (IF, representing the forest stage heavily invaded by Bai bamboo), bamboo was the only species in both the tree and shrub layers, with relatively few herbs.

### 2.3. Soil and Litterfall Sampling

We constructed five 5 × 5 m plots for each of the three forest stands. We sampled five replicates of each plot in an S-shape and mixed them into one sample. Within each of the plots, soil samples were collected from the 0–20 cm and 20–40 cm soil layers at five random points. The soil samples were separated into two and sieved through a 2 mm screen prior to measurement. To calculate the soil microbial biomass carbon (MBC), one fraction was kept at 4 °C. The second portion was air-dried to determine the soil's physicochemical properties and organic C fractions. Simultaneously, soil samples were taken using ring cutters from each of the two soil layers at each point. The soil samples were stored in sealed aluminum cans and brought back to the laboratory to determine the soil bulk density. We created 5 1 × 1 m small sample squares per forest stand and 15 sample squares in total. All the litter was collected from the sample squares and brought to the laboratory. Litterfall samples were dried to a constant mass and weighed to calculate biomass.

**Table 2.** Summary of vegetation species at different bamboo invasion stages.

| Stands | Lays | Species |
|---|---|---|
| BF | tree | *Adinandra megaphylla* Hu.<br>*Liquidambar formosana* Hance.<br>*Itea oblonga* Hand. |
| | shrub | *Ilex viridis* Champ. ex Benth.<br>*Ternstroemia gymnanthera* (Wight et Arn.) Beddome<br>*Lindera glauca* (Siebold & Zucc.) Blume<br>*Camellia oleifera* Abel.<br>*Oxyspora paniculata* (D. Don) DC.<br>*Maesa japonica* (Thunb.) Moritzi. ex Zoll.<br>*Eurya groffii* Merr. |
| | grass | *Curculigo orchioides* Gaertn.<br>*Arachniodes chinensis* (Rosenst.) Ching<br>*Alpinia psilogyna* D. Fang<br>*Acrophorus stipellatus* (Wall.) Moore<br>*Trachelospermum jasminoides* (Lindl.) Lem.<br>*Smilacina japonica* A. Gray<br>*Polygonum persicaria* L.<br>*Plagiogyria grandis* Copel. |
| MF | tree | *Adinandra megaphylla* Hu<br>*Itea oblonga* Hand<br>*Indosasa shibataeoides* McClure. |
| | shrub | *Camellia oleifera* Abel. |
| | grass | *Arachniodes chinensis* (Rosenst.) Ching<br>*Acrophorus stipellatus* (Wall.) Moore<br>*Trachelospermum jasminoides* (Lindl.) Lem.<br>*Smilacina japonica* A. Gray |
| IF | *tree* | *Indosasa shibataeoides* McClure. |
| | shrub | - |
| | grass | *Plagiogyria grandis* Copel.<br>*Alpinia psilogyna* D. Fang<br>*Arachniodes chinensis* (Rosenst.) Ching<br>*Acrophorus stipellatus* (Wall.) Moore |

## 2.4. Analytical Methods

The organic carbon of soil samples was calculated using the total organic carbon (TOC) analyzer (Muti N/C 3100, Analytik Jena, Jena, Germany). Soil pH was measured using a pH meter with a soil-to-water ratio of 1:2.5 (*w/v*). Soil alkaline hydrolyzed nitrogen (AN) was measured using the alkali dispersion method [23]. Soil available P (AP) was measured using 0.03 mol·L$^{-1}$ NH$_4$F and 0.025 mol·L$^{-1}$ HCl [24]. Soil available K (AK) was extracted using 1 mol·L$^{-1}$ ammonium acetate solution and determined using a flame photometer (FP6410 INESA, Shanghai, China).

## 2.5. Extraction and Analysis of the Soil Organic Carbon Fraction

SOC was categorized into five different fractions based on how organic carbon was classified [8]: easily oxidizable organic carbon (EOC), MBC, dissolved organic carbon (DOC), calcium-bound SOC (Ca-SOC), and iron/aluminum-bound SOC (Fe/Al-SOC). EOC was measured via oxidation with KMnO$_4$ [25]. DOC was extracted with K$_2$SO$_4$ [26]. MBC was determined via chloroform fumigation extraction [27], and the Ca-SOC and Fe/Al-SOC contents of the samples were determined by NaSO$_4$ and Na$_4$P$_2$O$_7$·10H$_2$O leaching, respectively [12].

### 2.6. Bacterial and Fungal Communities Analysis

After genomic DNA was extracted from the soil samples, the V3 and V4 regions of 16S rDNA were amplified using specific primers and barcodes [28]. The primer sequences were 341F: CCTACGGGGNGGCWGCAG; 806R: GGACTACHVGGGTATCTAAT. Purified amplification products (amplicons) were ligated at sequencing junctions to construct sequencing libraries, and Illumina sequencing was performed. After extracting genomic DNA from the soil samples, the ITS2 region of the ITS was amplified using specific primers with a barcode. The primer sequences were ITS3_KYO2: GATGAAGAACGYAGYRAA and ITS4: TCCTCCGCTTATTGATATATGC. The purified amplification products (amplicons) were connected to a sequencing connector, and a sequencing library was constructed and sequenced using Illumina [29].

### 2.7. Statistics and Analysis

Soil properties and SOC fractions were statistically analyzed using IBM SPSS (version 26.0; Chicago, IL, USA). One-way ANOVA was used to test for significant differences between soil samples, with $p \leq 0.05$ indicating significant differences. Data are expressed as mean $\pm$ standard deviation (SD).

The R package "microeco" was used for calculating the $\alpha$-diversity index and calculating principal coordinate analysis (PcoA) [30]. The R package "vegan" was used for calculating redundancy analyses (RDA) and to assess the influence of environmental factors on soil microbial communities. The R 4.3.1 software package "ran-domForest" was used to build a random forest model and identify key predictors of soil microbial communities [31]. The R software package "plmps" [32] was used to determine the effect of Bai bamboo invasion on SOC using partial least squares path modeling (PLS-PM).

## 3. Results

### 3.1. Soil Physical Properties and Available Nutrients

The soil pH of the broadleaved forest was lower than that of the mixed forest and increased in the Bai bamboo forest (Table 3, $p < 0.05$). The soil bulk density in the 0–20 cm soil layer significantly increased after bamboo invasion ($p \leq 0.05$) compared with broadleaved forest and mixed forest. Litterfall increased significantly with bamboo invasion ($p < 0.05$).

**Table 3.** Selected physical properties and litterfall biomass in different soil layers under different bamboo invasion stages.

| Soil Layer | Vegetation Type | pH (H$_2$O) | BD (g·cm$^{-3}$) | Litterfall Layer (g·m$^{-2}$) |
|---|---|---|---|---|
| | BF | 4.01 ± 0.03 a | 0.944 ± 0.06 b | 125.42 ± 25.89 c |
| 0–20 cm | MF | 3.80 ± 0.03 c | 1.04 ± 0.05 b | 352.25 ± 44.38 b |
| | IF | 3.91 ± 0.02 b | 1.326 ± 0.04 a | 444.62 ± 48.14 a |
| | BF | 4.51 ± 0.01 a | 1.328 ± 0.02 a | - |
| 20–40 cm | MF | 4.00 ± 0.02 c | 1.084 ± 0.11 c | - |
| | IF | 4.31 ± 0.09 b | 1.152 ± 0.11 a | - |

BD, soil bulk density; Litterfall, litterfall biomass. Different lowercase letters within a column indicate significant differences between the different bamboo invasion stages within the same soil layer, respectively, at the $p = 0.05$ level. Data are presented as the mean $\pm$ SD, $n = 10$.

The SOC, AN, AP, and AK contents of forest soils significantly differed among the different vegetation types—that is, bai bamboo forest, broadleaved forest, and mixed forest—with different soil depths ($p \leq 0.05$; Table 4). As the invasion of Bai bamboo into the broadleaved forest increased, the MBC, AN, and AK contents of the soil increased significantly in the upper soil layer ($p \leq 0.05$; Table 4).

**Table 4.** Selected chemical properties in different soil layers for different bamboo invasion stages.

| Soil Layer | Vegetation Type | SOC (g·kg$^{-1}$) | MBC/SOC (%) | AN (mg·kg$^{-1}$) | AP (mg·kg$^{-1}$) | AK (mg·kg$^{-1}$) |
|---|---|---|---|---|---|---|
| | BF | 41.23 ± 1.42 b | 0.99 ± 0.08 c | 277.17 ± 8.86 c | 3.67 ± 0.16 a | 65.45 ± 2.54 c |
| 0–20 cm | MF | 41.77 ± 1.59 b | 1.90 ± 0.15 b | 302.99 ± 56.55 b | 3.69 ± 0.84 a | 71.71 ± 0.87 b |
| | IF | 52.64 ± 1.71 a | 2.38 ± 0.23 a | 421.20 ± 27.17 a | 2.26 ± 0.70 b | 90.80 ± 0.02 a |
| | BF | 26.82 ± 1.05 a | 0.68 ± 0.17 b | 164.40 ± 7.44 b | 2.29 ± 0.04 a | 30.65 ± 5.39 a |
| 20–40 cm | MF | 22.19 ± 0.40 b | 1.17 ± 0.17 a | 164.40 15.49 b | 2.47 ± 0.29 a | 26.89 ± 0.53 c |
| | IF | 26.88 ± 1.01 a | 1.28 ± 0.32 a | 258.15 ± 34.64 a | 1.56 ± 0.08 b | 33.96 ± 20.8 a |

SOC, soil organic carbon; AN, available nitrogen; AP, available phosphorus; AK, available potassium; MBC/SOC, ratio of microbial biomass C to soil organic C. Different lowercase letters within a column indicate significant differences between the different bamboo invasion stages within the same soil layer, respectively, at the $p = 0.05$ level. Data are presented as means ± SD, $n = 10$.

### 3.2. SOC Fractions

The SOC fractions were significantly different among the three vegetation types ($p < 0.05$; Figure 1). As the invasion of Bai bamboo into the broadleaved forest increased, the soil MBC and DOC contents increased significantly ($p \leq 0.05$). While the soil EOC contents increased from broadleaved forest to mixed forest in the upper soil layer, the soil EOC contents of Bai bamboo forest and Bai bamboo invasion were lower than those of the mixed forest ($p \leq 0.05$). In the undersoil layers, the soil EOC content decreased and then increased ($p \leq 0.05$).

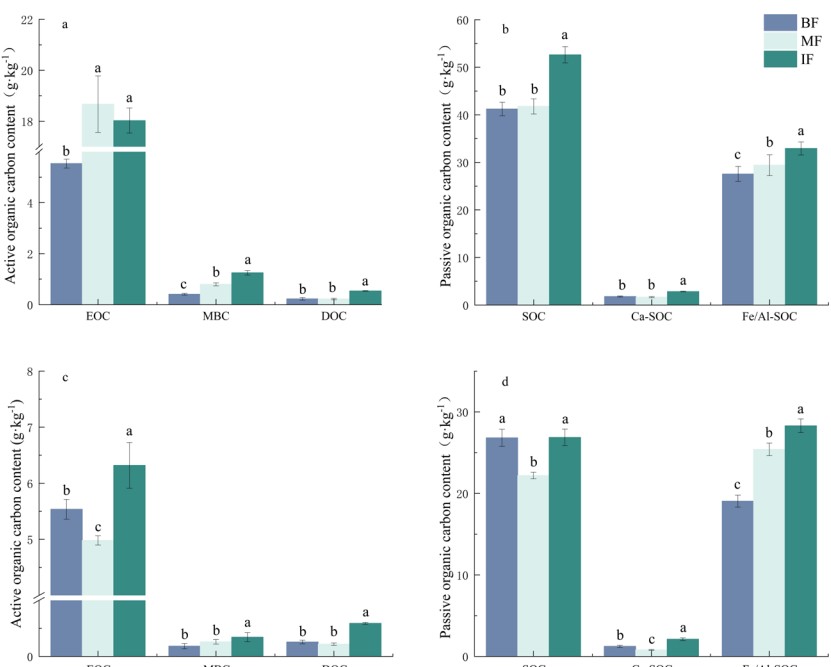

**Figure 1.** Active (**a**) and passive (**b**) SOC contents during the Bai bamboo invasion process in the 0–20 cm soil layer. Soil active (**c**) and passive (**d**) organic carbon contents during the bamboo invasion process in the 20–40 cm soil layer. Different lowercase letters indicate significant differences among different layers according to the independent samples *t*-test ($p \leq 0.05$). Error bars indicate standard deviation ($n = 10$).

In the 0–20 cm soil layer, the difference in the SOC content was not significant in the early period of the Bai bamboo invasion ($p > 0.05$). The SOC content increased during the complete Bai bamboo invasion stage ($p < 0.05$). With the invasion of Bai bamboo into the broadleaved forest, the Ca−SOC and Fe/Al−SOC contents increased ($p \leq 0.05$). The Ca−SOC and Fe/Al−SOC contents of the Bai bamboo forest were higher than those of the broadleaved forest in the 20–40 cm soil layer ($p \leq 0.05$). The difference in SOC content

between the Bai bamboo forest and broadleaved forest was not significant at 20–40 cm ($p > 0.05$).

The invasion of Bai bamboo significantly affected the soil's active organic carbon and passive carbon pool compositions. During the early period of Bai bamboo invasion, the EOC, MBC, and Fe/Al−SOC contents increased in the 0–20 cm soil layer ($p \leq 0.05$). The SOC and EOC contents significantly decreased in the 20–40 cm soil layer. As the vegetation was thoroughly invaded by Bai bamboo, the contents of the different C fractions increased significantly ($p \leq 0.05$).

### 3.3. Alpha Diversity of Soil Microbial Communities

The alpha diversity of microbial communities was evaluated using Chao1 and Shannon index assessments. Chao1 and Shannon indices of the bacterial community did not change significantly ($p > 0.05$) during Bai bamboo invasion in the 0–20 cm soil layer, while Chao1 and Shannon indices of the bacterial community increased significantly ($p \leq 0.05$, Figure 2) in the 20–40 cm soil layer. Chao1 and Shannon indices of fungal communities increased significantly ($p \leq 0.05$, Figure 2) during Bai bamboo invasion in the 0–40 cm soil layer.

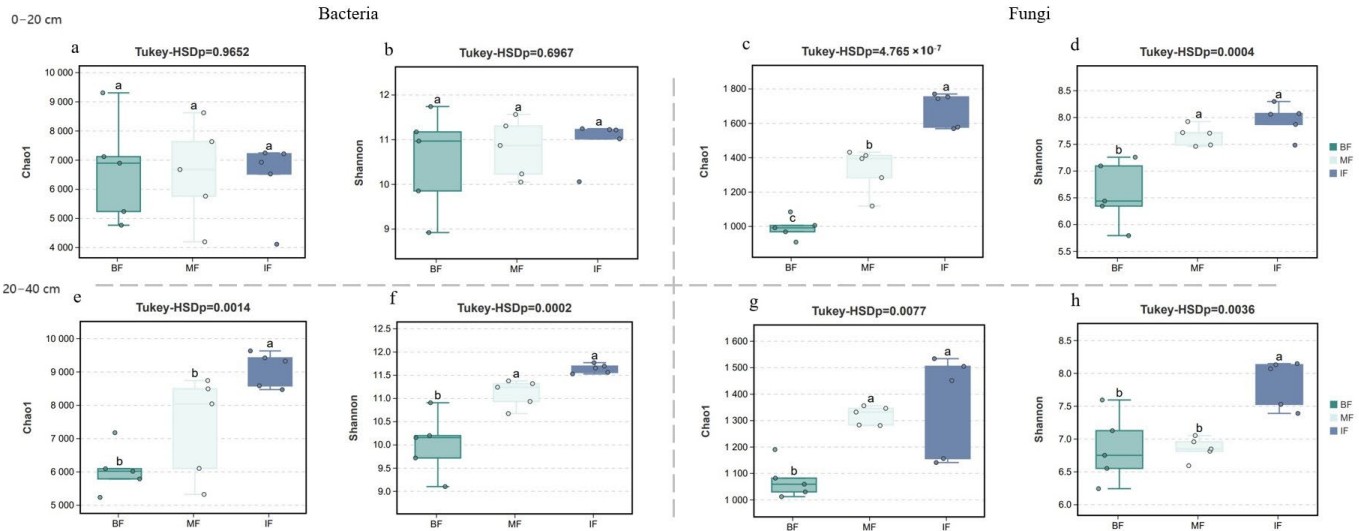

**Figure 2.** Alpha diversity indices of bacterial (**a**,**b**) and fungal (**c**,**d**) taxa for the soil samples from the 0–20 cm soil layer. Alpha diversity indices of bacterial (**e**,**f**) and fungal (**g**,**h**) taxa for the soil samples from the 20–40 cm soil layer. Different lowercase letters within a column indicate significant differences between the different bamboo invasion stages within the same soil layer, respectively, at the $p = 0.05$ level.

### 3.4. Changes in the Structure of Bacterial and Fungal Communities

Acidobacteria was the most abundant phylum from the broadleaved and mixed forests. The relative abundance of Acidobacteriota accounted for over 20% of the total sequences in the 0–20 cm soil layer. Firmicutes accounted for the second highest percentage of the total sequences, which was almost 50% in the 20–40 cm soil layer samples of the broadleaved forest soil. These were followed by Actinobacteria, Chloroflexi, and Proteobacteria, accounting for approximately 10%, 8.81%, and 7.69% of the total sequences, respectively. With the invasion of broadleaved forest by Bai bamboo, the relative abundances of Acidobacteria and Actinobacteria increased in the 20–40 cm soil layer, while that of Firmicutes decreased.

Within the fungal communities, the most abundant fungi were Ascomycota, accounting for 53.56%, followed by Basidiomycota, Mortierellomycota, Mucoromycota, and Rozellomycota, accounting for 25.02%, 9.41%, 3.24%, and 3.07% in the 0–20 cm and 20–40 cm soil layers, respectively.

At the genus level, the abundance of *Kitasatospora* increased significantly ($p \leq 0.05$). Meanwhile, those of *Tumebacillus*, *Candidatus_Solibacter*, *Candidatus_Koribacter*, and

*Acidothermus* significantly decreased ($p \leq 0.05$) in the 0–20 cm soil layers with Bai bamboo invasion (Figure 3). Additionally, the abundances of *Kitasatospora* and *Candidatus_Koribacter* increased ($p \leq 0.05$) in the 20–40 cm soil layers. The abundances of *Acidothermus*, *Candidatus_Koribacter*, and *Candidatus_Udaeobacter* decreased significantly ($p \leq 0.05$) in the 20–40 cm soil layers.

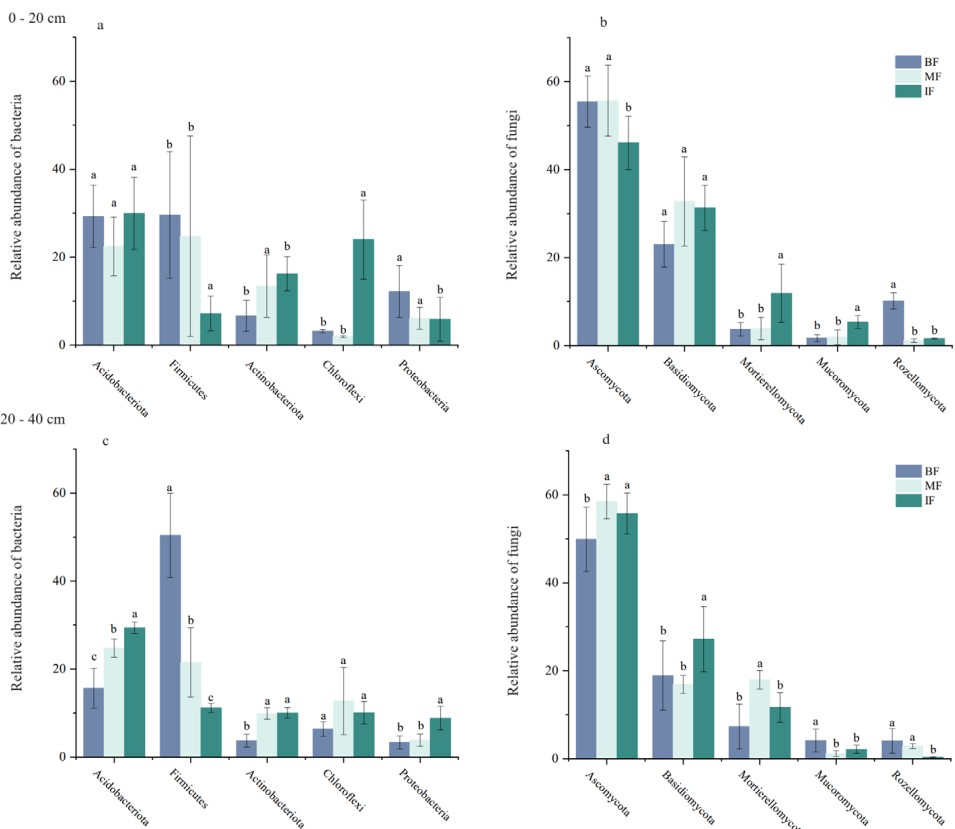

**Figure 3.** Bacterial and fungal compositions at the phylum level in different soil layers ((**a**,**b**) 0–20 cm and (**c**,**d**) 20–40 cm) during Bai bamboo invasion in broad-leaved forests. Average relative abundances of the top five bacterial and fungal phyla. Different lowercase letters within a column indicate significant differences between the different bamboo invasion stages within the same soil layer, respectively, at the *p* = 0.05 level.

In the 0–20 cm soil layer, the abundance of the fungus *Hygrocybe* increased significantly (*p* < 0.05). The abundances of *Mortierella*, *Hygrocybe*, *Pseudohydnum*, and *Chloridium* increased significantly ($p \leq 0.05$) in the 20–40 cm soil layers. The abundances of *Trichoderma*, *Saitozyma*, *Clitopilus*, and *Apiotrichum* decreased significantly ($p \leq 0.05$).

PcoA demonstrated differences in microbial communities at different invasion stages (Figure 4). Bacteria and fungi accounted for 35.51% and 23.21% of the pcoa1 and pcoa2 variation, respectively. In terms of bacterial and fungal community composition, three distinct communities were all formed in three invasion period samples from both the upper and lower soil layers. This shows that the invasion of Bai bamboo into the broadleaved forest caused bacterial and fungal community shifts.

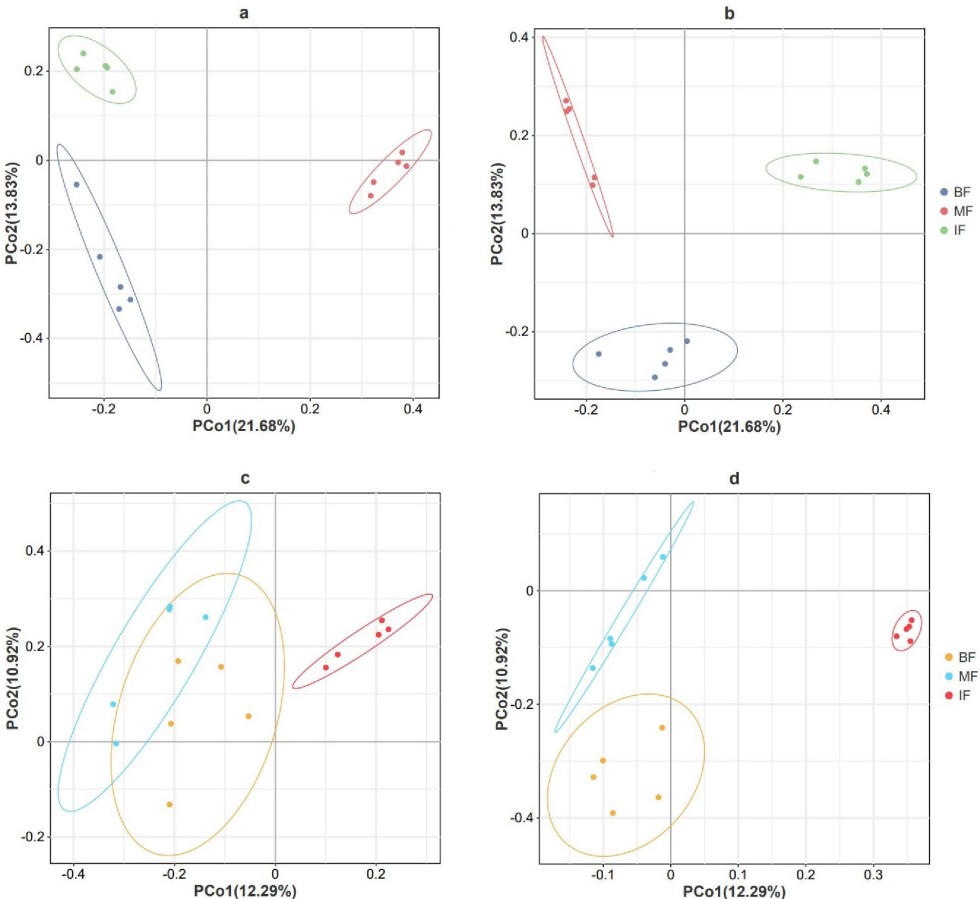

**Figure 4.** Principal coordinate analysis of soil bacterial (**a**,**b**) and fungal (**c**,**d**) communities in different soil layers (0–20 cm and 20–40 cm) during Bai bamboo invasion in broadleaved forests.

### 3.5. Correlations between Soil Environmental Factors and Bacterial and Fungal Community

We conducted RDA and Mantel tests to evaluate the relationships between soil factors and microbial community structure (Figure 5a,b). RDA indicated that RDA1 and RDA2 explained 36.71% and 27.42% of the variation in the bacterial communities, respectively (Figure 5a). Soil environmental factors, including the MBC, Ca-SOC, Fe/Al-SOC, DOC, AN, Ph, SBD, and litterfall contents, were significantly correlated with bacterial communities ($r^2$ = 0.298, 0.569, 0.511, 0.810, 0.328, 0.356, 0.304, 0.608, and 0.888, respectively). Within the fungal communities, RDA1 and RDA2 explained 29.61% and 18.85% of the variance (Figure 5b). Soil chemical parameters, including MBC, SOC, EOC, Ca-SOC, Fe/Al-SOC, DOC, AN, AP, AK, SBD, and litterfall content, were significantly correlated with fungal communities ($r^2$ = 0.568, 0.412, 0.229, 0.810, 0.829, 0.335, 0.736, 0.603, 0.297, 0.376, 0.609, and 0.245, respectively).

Random forest analysis was also conducted to identify the main soil factors responsible for bacterial communities (Figure 6). Fe/Al-SOC was the most important factor in bacterial communities (percentages of increased mean square error (%IncMSE: 7.53%), followed by Ph, DOC, and AP (%IncMSE: 6.88%, 5.63%, and 4.92%, respectively). Within the fungal communities, EOC was the most important factor (%IncMSE: 12.77%), followed by Fe/Al-SOC, Ca-SOC, litterfall, and AP (%IncMSE: 9.98%, 7.03%, 6.25%, and 5.82%, respectively).

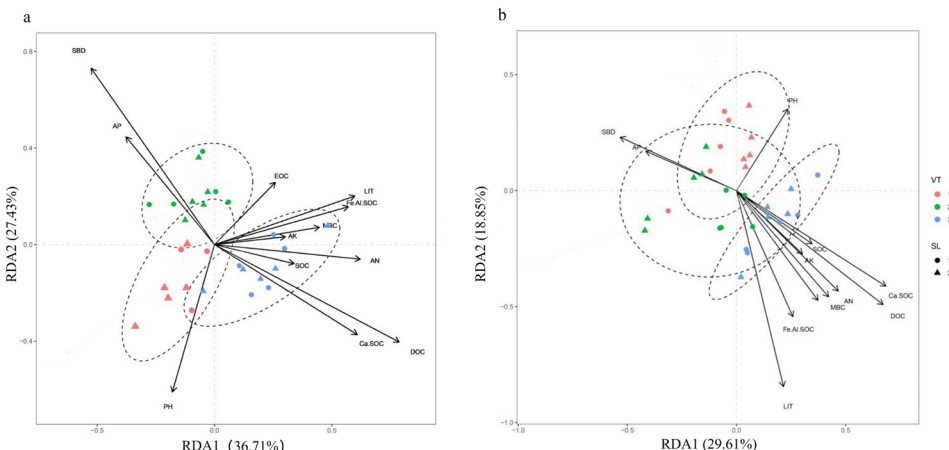

**Figure 5.** Redundancy analysis (RDA) of bacterial (**a**) and fungal (**b**) communities and soil environmental factors in different soil layers (0–20 cm and 20–40 cm) during Bai bamboo invasion in broad-leaved forests.

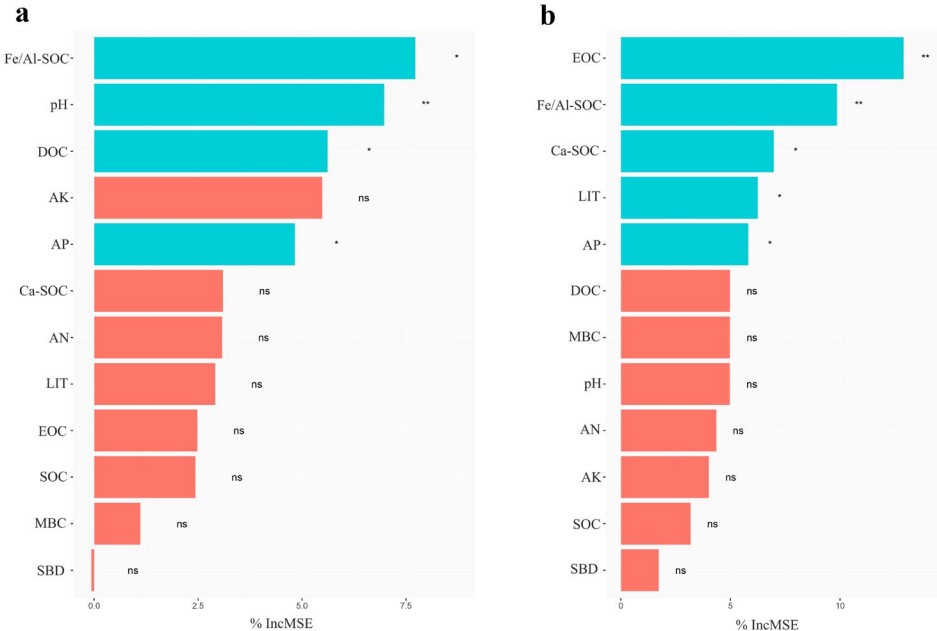

**Figure 6.** Random forest analysis revealed the relative contributions of soil environmental factors in determining soil bacterial (**a**) and fungal (**b**) communities in different soil layers (0–20 cm and 20–40 cm) during Bai bamboo invasion in broadleaved forests. The bacterial and fungal community data represent the α diversity index and the relative abundance of keystone taxa. ** $p \leq 0.01$; * $p \leq 0.05$; ns, $p > 0.05$. %IncMSE: percentage of increased mean square error.

### 3.6. Factors Driving Soil Organic Carbon

The PLS-PM results showed that SOC was directly affected by litterfall biomass (Figure 7, path coefficient = 0.51) and fungal community (Figure 7, path coefficient = 0.69), respectively. Litterfall biomass had an indirect effect on the fungal community by altering the available nutrients (path coefficient = 0.68) and soil physical properties (path coefficient = 0.21).

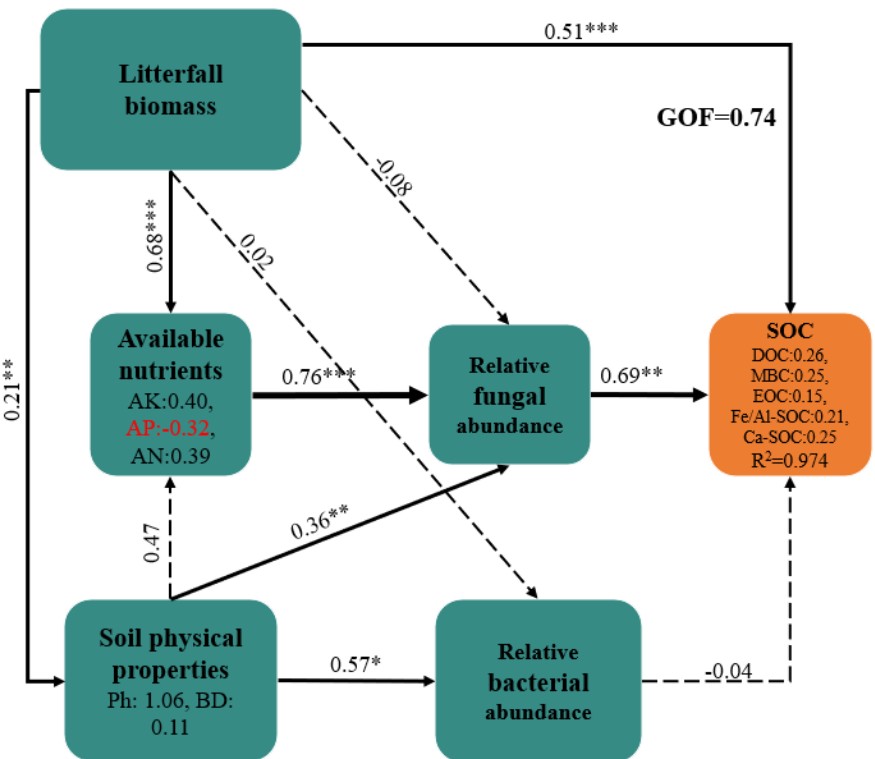

**Figure 7.** Partial least squares path modeling (PLS−PM) results show direct and indirect effects of Bai bamboo invasion; rectangular boxes indicate variables included in the model. The black arrows indicate significant ($p < 0.05$) positive and negative correlations, whereas black dotted lines indicate non-significant ($p > 0.05$) relationships. The red letters indicate that the variable and the factor represented by the square box are negatively correlated. Numbers adjacent to arrows represent standardized path coefficients. The path width is proportional to the path coefficient. Additionally, the R-square values associated with the SOC variable indicate the percentage of variance explained by the relationships with other variables. GOF, goodness of fit. *** $p < 0.001$, ** $p < 0.01$, * $p < 0.05$.

## 4. Discussion

### 4.1. Effects of Bamboo Invasion on Litterfall Biomass

In our study, the invasion of Bai bamboo into broadleaf forests led to an increase in litterfall biomass per unit area of karst forests (Table 3). Some studies found that Moso bamboo invasion in broadleaf forests reduced plant diversity and litterfall biomass [33]. And the effect of bamboo invasion of neighboring forests on litterfall mass varies depending on the type and location of the forest [34]. This opposite phenomenon is due to differences in the growth of pendulum bamboos in mixed and pure forests; during the mixed bamboo-broad forest (MF) period of broadleaf forests, increases the density of trees that provides more litterfall biomass per unit area (Table 2). During the invaded Bai bamboo forest (IF) period, the average plant height of Bai bamboo was reduced by about 35%, and the average DBH was reduced by about 33%, while the plant density was increased by almost 92%. More plant density increases the biomass of litterfall biomass per unit area of the forest system [35]. Bai bamboo changed its growth strategy after the invasion into broadleaf forests by increasing plant density and decreasing plant height, which led to an increase in the litterfall biomass per unit area (Table 2).

### 4.2. Effects of Bamboo Invasion on Soil Organic Carbon

Previous studies indicated that bamboo invasion into broadleaf forests had a great impact on C fractions [6]. Our results showed that SOC content in the 0–20 cm soil layer increased significantly after the replacement of the broadleaf forest by the Bai bamboo forest. In the MF (Figure 1b), the SOC alteration was not significant in the 0–20 cm soil

layer ($p > 0.05$), and in the 20–40 cm soil layer, the content of SOC and EOC in MF was lower than BF (Figure 1c,d), which was due to the fact that EOC was more available to the plant compared to other carbon fractions, and the growth of bamboo consumed the EOC, AN, and AK in the early stage of invasion [36]. As the Bai bamboo invasion, the increase of litterfall biomass, the soil-available nutrients, the AN and AK contents increased significantly in MF and IF, while the AP content decreased significantly in IF (Table 4). The changes in AP were similar to those in soils invaded by Moso bamboo [37], which may be due to the increase in density of bamboo after the complete invasion of broad-leaved forests by Bai bamboo and the growth of bamboo roots and sticks consumed a large amount of AP in the soil [38]. The changes in AK and AN were the opposite of those in the Moso bamboo invasion, which was due to the increase in litterfall biomass. Litterfall is an important source of soil nutrients, and litterfall and decomposition return nutrients to the soil [39].

The MBC is commonly used to evaluate soil microorganisms [40]. It is directly involved in soil biochemical transformation processes [41]. With the invasion of broad-leaved forest by Bai bamboo, the MBC/SOC ratio increased significantly (Table 4). This suggests that Bai bamboo promotes soil carbon cycling efficiency in broad-leaved forests. The combination of Fe/Al and Ca with soil organic carbon can improve SOC fixation and thus improve SOC library stability [42]. Our study showed that Fe/Al-SOC is about 15 times more abundant than Ca-SOC, and Fe/Al oxides are higher in karst soils (Figure 1b,d). With the invasion of Bai bamboo, the Fe/Al-SOC and Ca-SOC contents of the passive organic carbon pool increased significantly in all soil layers (Figure 1b,d). This suggested that the invasion of Bai bamboo promoted the accumulation of inert carbon in karst soils; it favored the sequestration of organic carbon in soil under karst forest.

### 4.3. Effects of Bamboo Invasion on Bacterial and Fungal Communities

For the first time, we studied the microbial community of the soil of the Bai bamboo forest in the karst area. The predominant bacterial species in the soil of the Bai bamboo forest was Acidobacteria at the phylum level of bacteria. This is roughly similar to the dominant bacterial flora in the soil of the Moso bamboo forest and the Lei bamboo forest [43]. The two fungi with the highest relative abundance at the fungal phylum level were Ascomycota and Basidiomycota, which were also common dominant species in the soil of the Moso bamboo forest [37]. At the same time, we found that the Bai bamboo invasion changed the soil bacterial and fungal community composition. PcoA showed that soil samples from three different stages of the Bai bamboo invasion formed two different clusters in the analysis of bacterial and fungal community composition (Figure 5a,b).

There were differences in the effects of bamboo invasion on the diversity indices of bacterial and fungal communities, and the chao1 and Shannon (Figure 2a,b) indices of bacterial communities did not differ significantly before and after the invasion of Bai bamboo. This is consistent with the findings of Liu [44]. In our study, the relative abundances of Actinobacteriota and Chloroflexi had increased in the 0–20 cm soil layer while the relative abundances of Acidobacteria and Proteobacteria increased in the 20–40 cm soil layer significantly (Figure 3a). Previous studies found that the relative abundance of Actinobacteria and Acidobacteria increased with increasing soil-available nutrients [45], which is consistent with the results. Proteobacteria can promote the increase of organic carbon fractions by improving C-efficient and respiration, while Acidobacteria are able to utilize a variety of carbohydrates as carbon sources [46]. Chloroflexi is able to participate in the decomposition of sugars and plant-derived compounds and fix $CO_2$ through the Wood–Ljungdahl pathway [47]. The increase in the relative abundance of Actinobacteria and Acidobacteria could promote the decomposition of litterfall biomass [48]; the bamboo invasion had caused a shift in the structure of the bacterial community in karst forest soils, accelerating the efficiency of the conversion of organic matter to SOC.

Bai bamboo invasion increased the diversity (Chao1 and Shannon indices) of fungal communities in the 0–40 cm soil layer (Figure 2c,d,g,h). Our study findings demonstrated that the change in forest SOC content from bamboo invasion was realized through changes

in the fungal community structure. Fungi can improve karst soil nutrients by promoting the release of nutrients from litterfall biomass, thereby increasing SOC and available nutrient content [47,48].

Our study found a significant decrease in the relative abundance of the bacterium *Tumebacillus* and an increase in the relative abundance of the fungus *Mortierella* with the invasion of Bai bamboo in the 0–20 cm soil layer (Table 5, *p* < 0.05). *Tumebacillus* and *Mortierella* have a strong ability of hydrolysis; *Mortierella* abundance increased in more humus-rich environments, and the increase in litterfall in IF stands created a more suitable environment for *Mortierella* [49,50]. The relative abundances of the soil phyla Mortierellomycota and Basidiomycota increased significantly during the late stage of the Bai bamboo invasion (Figure 3b,d). This is consistent with the findings of Liu, CX [37]. Some studies have demonstrated that the efficiency of organic carbon storage is significantly higher in soils enriched with Mortierellomycota flora [51]. Mortierellomycota and Basidiomycota can utilize carbon sources composed of cellulose and lignin because of their ability to secrete various cellulose- and hemicellulose-degrading enzymes [52,53]. This demonstrates that the alteration of fungal community structure by the invasion of Bai bamboo could, in turn, promote the decomposition of litterfall. Fungi are essential for SOC formation, especially for the more stable Fe/Al-SOC and Ca-SOC [54]. The increase in EOC, Fe/Al-SOC, and Ca-SOC could promote the change of fungal community structure towards the capacity of litterfall decomposition (Figure 6b). These prove that, after the Bai bamboo invasion, the capacity of the fungal communities to break down lignin and cellulose increased. Additionally, fungal communities were indirectly affected by litterfall via changing soil-available nutrients (AN, AP, and AK) and soil physical properties (pH and BD).

**Table 5.** Relative abundance of major microbiological genera in the forest soil in different Bai bamboo invasion stages.

| | Phylum | Genus | 0–20 cm | | | *p*-Value | 20–40 cm | | | *p*-Value |
|---|---|---|---|---|---|---|---|---|---|---|
| | | | BF | MF | IF | | BF | MF | IF | |
| Bacteria | Firmicutes | *Tumebacillus* | 16.24 | 11.11 | 3.61 | 0.001 | 4.59 | 32.75 | 11.86 | 0.181 |
| | Firmicutes | *Bacillus* | 9.12 | 7.70 | 1.83 | 0.373 | 2.07 | 7.23 | 4.49 | 0.28 |
| | Chloroflexi | *HSB_OF53-F07* | 1.09 | 1.87 | 8.63 | 0.146 | 4.43 | 2.95 | 6.47 | 0.13 |
| | Proteobacteria | *Burkholderia-Caballeronia-Paraburkholderia* | 4.84 | 3.43 | 1.56 | 0.052 | 2.61 | 0.99 | 0.80 | 0.28 |
| | Actinobacteriota | *Acidothermus* | 1.74 | 2.26 | 2.30 | 0.044 | 1.41 | 0.79 | 1.21 | 0.048 |
| | Actinobacteriota | *Kitasatospora* | 0.37 | 0.49 | 3.14 | 0.001 | 3.06 | 0.71 | 1.63 | 0.001 |
| | Acidobacteriota | *Candidatus_Solibacter* | 2.02 | 1.95 | 1.24 | 0.024 | 0.91 | 0.40 | 1.02 | 0.08 |
| | Acidobacteriota | *Candidatus_Koribacter* | 1.93 | 1.59 | 1.39 | 0.02 | 0.52 | 0.25 | 0.69 | 0.001 |
| | Verrucomicrobiota | *Candidatus_Udaeobacter* | 0.84 | 0.38 | 0.73 | 0.817 | 1.66 | 0.94 | 0.41 | 0.001 |
| | Proteobacteria | *Acidibacter* | 0.95 | 1.12 | 0.47 | 0.116 | 1.02 | 0.39 | 0.27 | 0.036 |
| Fungi | Mortierellomycota | *Mortierella* | 3.67 | 3.90 | 11.88 | 0.063 | 7.33 | 17.97 | 11.67 | 0.001 |
| | Ascomycota | *Trichoderma* | 10.79 | 5.18 | 3.41 | 0.252 | 6.67 | 24.48 | 2.12 | 0.001 |
| | Basidiomycota | *Saitozyma* | 9.61 | 5.60 | 2.01 | 0.072 | 6.66 | 5.86 | 1.93 | 0.001 |
| | Basidiomycota | *Hygrocybe* | 0.25 | 0.30 | 13.18 | 0.001 | 1.68 | 0.17 | 3.25 | 0.001 |
| | Basidiomycota | *Pseudohydnum* | 0.21 | 12.77 | 0.23 | 0.972 | 0.21 | 1.46 | 0.92 | 0.02 |
| | Mucoromycota | *Umbelopsis* | 1.30 | 0.99 | 3.49 | 0.216 | 2.53 | 0.41 | 0.99 | 0.258 |
| | Basidiomycota | *Clitopilus* | 3.58 | 0.47 | 0.59 | 0.198 | 3.96 | 0.19 | 0.41 | 0.011 |
| | Ascomycota | *Chloridium* | 0.45 | 1.10 | 0.71 | 0.737 | 0.78 | 3.57 | 1.13 | 0.001 |
| | Ascomycota | *Talaromyces* | 1.62 | 1.25 | 1.43 | 0.742 | 1.10 | 0.90 | 1.29 | 0.353 |
| | Basidiomycota | *Apiotrichum* | 1.80 | 0.99 | 0.40 | 0.184 | 0.90 | 2.72 | 0.38 | 0.001 |

### 4.4. Mechanisms Driving Soil Organic Carbon by Bamboo Invasion

In our study, the SOC content increased with the bamboo invasion of broadleaved forest. The results of PLS-PM suggested that litterfall and fungal communities had direct and positive effects on SOC. The fungal communities linked to C cycling, rising EOC,

Fe/Al-SOC, and Ca-SOC promoted shifts in fungal community structure. Additionally, the fungal community in pure Bai bamboo forest soil was able to decompose litterfall more efficiently. Interactions between litterfall and fungal communities led to the accumulation of more SOC in pure Bai bamboo forest.

Litterfall input was the main driver of soil organic carbon changes in broadleaf forests invaded by Bai bamboo. Our research shows that the Bai bamboo invasion increased the litterfall biomass in karst forests. Litterfall could leach large amounts of nitrogen, phosphorus, and potassium through rainfall [55]. Meanwhile, with an increase in litterfall biomass, bamboo invasion increased soil nitrification through bacterial nitrifying capacity and soil urease activity [56]. Previous research has illustrated that dissolved organic matter produced by litter changes the soil microbial community [57]. Our study found a mechanism for the effect of litterfall on microorganisms; that is, the effect of litterfall on microorganisms was realized by changing the soil's available nutrients (Figure 7).

## 5. Conclusions

In the karst region, we found that native bamboo invasion of broad-leaved forests had a significant influence on SOC; bamboo invasion directly affects SOC through increasing litterfall biomass and altering fungal community structure. Bamboo invasion atlered the litterfall biomass (Figure 8), which increased the soil-available nutrients (AN and AK). The changes in soil-available nutrients and soil physical properties promoted the transformation of microbial community structure. Our study found that SOC was mainly influenced by soil fungal communities rather than bacteria. Compared with bacteria, changes in fungal community structure received a greater influence from litterfall biomass, and soil fungi shifted to a community structure more conducive to litterfall as decomposition and SOC sequestration with the Bai bamboo invasion. In summary, there was a significant enhancement in the SOC sequestration capacity under multiple uses of litterfall by fungal communities. Therefore, we suggest that the native bamboo invasion into broadleaf forests can increase the carbon sink potential of karst forests and, at the same time, through more research studies on trees and soils in karst regions. Moreover, future research is warranted to quantify seasonal changes in litterfall biomass at different invasion stages.

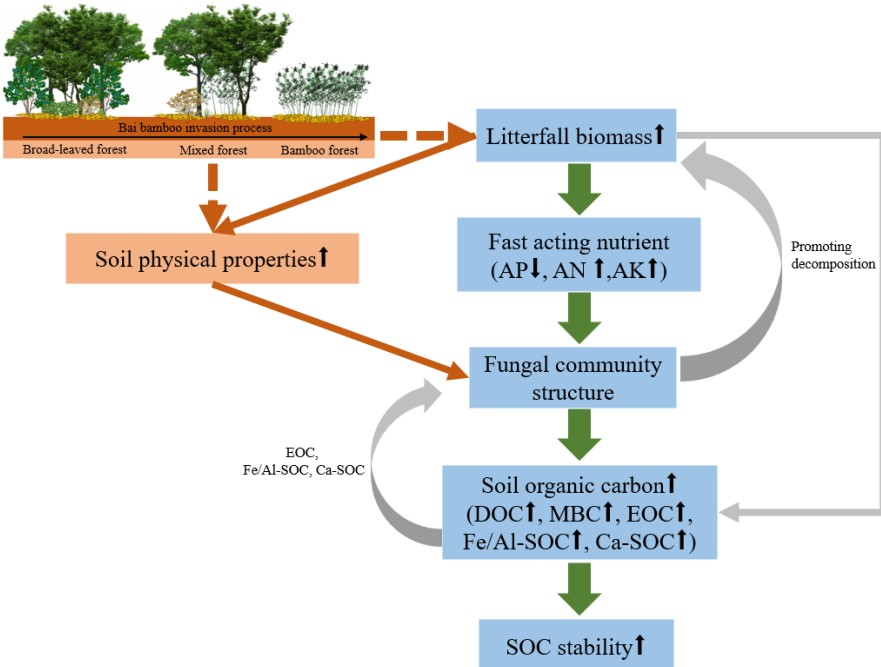

**Figure 8.** Conceptual diagram illustrating the mechanisms of Bai bamboo invasion of broadleaved forests in increasing soil C sequestration. The upward and downward dark arrows indicate increases and decreases in processes or values, respectively, during stand development.

**Author Contributions:** Z.C.: conceptualization, investigation, visualization, writing—original draft, writing—review and editing; X.Z.: investigation, supervision, resources, writing—review and editing; X.X.: writing—review and editing; Z.W.: writing—review and editing, funding acquisition; Z.H.: writing—review and editing; G.G.: writing—review and editing; J.Z.: writing—review and editing. All authors have read and agreed to the published version of the manuscript.

**Funding:** This research was supported by the Fundamental Research Funds of CAF: CAFYBB2022XE002, and the People's Government of Zhejiang Province−Chinese Academy of Forestry cooperative project: 2023B04.

**Data Availability Statement:** Not applicable.

**Acknowledgments:** The authors thank the Forestry Bureau of Long Sheng County, Guilin City, for their help during the field sampling process.

**Conflicts of Interest:** The authors declare no conflict of interest.

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
