# Peer review of "Native Bamboo (Indosasa shibataeoides McClure) Invasion of Broadleaved Forests Promotes Soil Organic Carbon Sequestration in South China Karst"

_forests, doi:10.3390/f14112135_

Round 1
Reviewer 1 Report
Comments and Suggestions for Authors
Dear Authors,
I read your manuscript entitled " Native bamboo (Indosasa shibataeoides McClure) invasion of broadleaved forests promotes soil organic carbon sequestration in South China karst". It is really an interesting topic in the ecology and management of forest soil and mitigating climate change. I enjoy reading your work. In my point of view, it is a comprehensive research about bamboo invasion in karst forest. Of course, the conclusion needs confirmation by further studies. I left some comments in the text body to improve your work.
All the best,
Reviewer

In my point of view, the quality of language is good.
Reviewer 2 Report
Comments and Suggestions for Authors
Currently, the problems of assessing the carbon pool of soils and the influence of phyto-edaphic factors are becoming urgent due to climate change on the planet. In this regard, the proposed work is very relevant, since currently the invasion of various plants can disrupt the natural rhythms of the cycle of biophilic elements, the strut-functional organization of the microbial community of soils. This is especially true in the southern territories, where the processes of decomposition of organic matter occur quite intensively, and therefore the effect of changes in the phytocomplex on soils can manifest itself much more quickly compared to the northern territories, for example, the boreal belt. The work looks very interesting, it will be necessary for specialists in various fields, especially soil scientists and microbiologists. The materials are presented clearly, the tables are "readable" easily. I hope that the scientific community will appreciate this work, especially since the terms are clear and used in the right context. There are comments in the work, which are presented below.
17 - a big request to specify the soil according to WRB
41- provide data on the chemical composition of this plant, what is its peculiarity. Please strengthen this part a little with research in the field of biochemistry/chemistry.
36-86- it is written simply, it is necessary to "strengthen" this part! provide data on the role of microorganisms in the transformation of organic matter, there may be work in the field of molecular interaction, and it is also possible to point out in more detail the changes in the soil-absorbing complex in violation of the structure of the microbiocenosis.
36- write down what karst forests are, many do not know in the world community
87-93 - very well written, everything is clear, thank you
108-119- you have well written in 16-17...we need to start with this.
109-specify the WRB soils that are on the plots.This is necessary for understanding process diagnostics.
156-there are no references to the works
figure 1-3,4 ,6 is not readable, a unread illustration. Can separate them?, especially since there are no big restrictions
292 - write... were significantly correlated 292 with fungal communities (r2 = 0.568, 0.412, 0.229, 0.810, 0.829, 0.335, 0.736, 0.603, 0.297, 293 0.376, 0.609, and 0.245, respectively)...something's wrong...please specify..its change from 0 to 1...
figure 6- remove the gray background, the text is not clearly visible
figure 7 - not clearly gray lines, make dotted lines and black, I recommend
342 - the link is not clear, does it coincide with them or is it the author's?
356 - a word with a capital letter, for what?
375 - I wanted to note that Tumebacillus and Mortierella they are strong hydrolytics, are part of the active group of the cellulolytic complex of the soil microbiota. These microorganisms are common in the soils of the northern territories (taiga zone of the European north). In this regard, I would like you to focus on these microorganisms, especially since Tumebacillus sharply reduces the number, and Mortierella increases. What about Bacillus, they are cosmopolitans, they are everywhere! Why is there such a change? what do you think?
440- it's not worth starting the conclusion with a drawing (the drawing is very revealing, good, thank you)
Once again I want to thank you for an interesting article, very important. Thank you. Good luck in your scientific work!
Comments on the Quality of English LanguageThe English language is clear, all terms are defined precisely. The sentences are small, but they are built correctly, the meaning is clear. Everything is clear. I think scientists from other countries will understand everything!
